# Treatment of Recurrent Giant Cell Tumor of Bones: A Systematic Review

**DOI:** 10.3390/cancers15133287

**Published:** 2023-06-22

**Authors:** Charalampos Pitsilos, Panagiotis Givissis, Pericles Papadopoulos, Byron Chalidis

**Affiliations:** 12nd Orthopaedic Department, Aristotle University of Thessaloniki, 54635 Thessaloniki, Greece; xnostos@hotmail.com (C.P.); perpap@otenet.gr (P.P.); 21st Orthopaedic Department, Aristotle University of Thessaloniki, 57010 Thessaloniki, Greece; pgivissis@gmail.com

**Keywords:** giant cell tumor, second recurrence, treatment, recurrence rate, adjuvant therapy, curettage, en-bloc resection

## Abstract

**Simple Summary:**

There is limited evidence in the current literature about the best management of the recurrent giant cell tumor of bones (GCTB). The aim of this systematic review is to summarize and analyze the contemporary treatment modalities of recurrent GCTB and evaluate their impact on second recurrence rate and function. Based on our findings, the combination of intralesional curettage and cavity filling with polymethylmethacrylate cement yields the best results and outcomes.

**Abstract:**

The giant cell tumor of bones (GCTB) is a benign bone tumor with high postoperative recurrence potential. No specific treatment protocol has been developed to date in case of tumor recurrence, and the kind of re-operative surgery depends upon the surgeon’s preferences. The aim of this systematic review is to determine the second recurrence rate and the respective functional results of the available treatment options applied to recurrent GCTB. Medline/PubMed and Scopus were searched to identify articles published until March 2023. Twelve studies fulfilled the inclusion criteria, comprising 458 patients suffering from recurrent GCTB. The overall incidence of second recurrence was 20.5%, at a mean interval of 28.8 months after the first surgery, and it was more evident after intralesional curettage (IC) surgery than en-bloc resection (EBR) (*p* = 0.012). In the IC group of patients, the second recurrence rate was lower and the functional outcome was greater when polymethylmethacrylate cement (PMMAc) was used as an adjuvant instead of bone grafting (*p* < 0.001 for both parameters). Reconstruction of the created bone defect after EBR with a structural allograft provided a better outcome than prosthesis (*p* = 0.028). According to this systematic review, EBR (first choice) and IC with PMMAc (second choice) are the best treatment options for recurrent GCTB.

## 1. Introduction

Giant cell tumor of bones (GCTB) is a primary benign but locally aggressive bone tumor, most commonly occurring in long bones of young adults [1]. It accounts for approximately 4–10% of all primary bone tumors and 15–20% of all benign bone tumors. The most frequently affected site is around the knee joint, followed by the distal radius and the proximal humerus [2].

The mainstay of treatment is surgical removal of the tumor. Based on the extent of the local disease, the surgical options vary from intralesional curettage (IC) to en-bloc resection (EBR) [3]. In most cases, the IC is combined with local adjuvants, including alcohol, phenol, liquid nitrogen, argon beam coagulation and thermo- or cryo-ablation [4,5]. Additionally, the remaining cavity after IC is usually filled with autologous or allogenous bone graft (BG) or polymethylmethacrylate cement (PMMAc) [6]. On the other hand, EBR is followed by reconstruction of the created bone defect. The selection of the reconstructive method depends on tumor location, available bone stock and integrity of articular surfaces [7,8]. A bone graft (structural autograft or allograft) is used in case of joint preservation or joint fusion. Alternatively, a reconstruction prosthesis is applied to restore joint mobility and function [9]. Radiotherapy, bisphosphonates and denosumab are recommended for nonresectable lesions but may also supplement surgical therapy [10].

Local recurrence (LR) after surgery is not uncommon and may be encountered in up to 53% of cases [11]. It is generally accepted that LR is precipitated by the postoperative residual tumor tissue, which might be present either in the remaining non-resected bone or surrounding soft tissues [12]. The EBR is associated with decreased relapse rate compared to IC due to the aggressive and extensive debridement of the pathological bone tissue [13]. However, the IC is preferable, as EBR has been associated with poor functional results [14]. Currently, there are no prognostic factors that can reliably determine the risk for LC [15]. However, the tumor location in long tubular bones, the increased number of surgical interventions, the occurrence of cortical destruction with extracompartmental spread, the existence of histological criteria for malignancy and the presence of p53 and cyclin D1 expression in mononuclear tumor cells and giant multinuclear cells, respectively, have been considered predisposing factors for higher LR rate [16].

There is also some evidence that the second recurrence may be quite common after surgical treatment of primary GCTB [17]. Despite this finding, many physicians follow the same treatment protocol for recurrent lesions, and IC is re-applied in a similar manner as in the initial operative procedure [18]. On the contrary, other surgeons and oncologists suggest a more aggressive approach to tumor recurrence and support EBR instead of IC, despite the risk of functional deficit [19,20,21].

While GCTBs have been widely investigated in the primary setting, recurrent tumor cases have not been correspondingly studied and analyzed. This systematic review aims to present the current evidence regarding the effectiveness of the available treatment options for the management of recurrent GCTB.

## 2. Materials and Methods

### 2.1. Study Type

A systematic review was conducted under the Preferred Reporting Items for Systematic Review and Meta-Analysis (PRISMA) guidelines.

### 2.2. Data Sources and Searches

Two electronic databases were used to develop this systematic review: Medline/PubMed and Scopus. These databases were combed for papers published until March 2023. Following Title and Abstract screening, the keywords that were used to obtain eligible articles were “Giant cell tumor” AND “bone” AND “recurrence” OR “recurrent” AND “treatment”.

### 2.3. Eligibility Criteria and Study Selection

Studies about the treatment of recurrent GCTB were eligible for further analysis. The inclusion criteria comprised studies published in English language journals during the past 20 years and contained at least 5 patients treated for recurrent GCTB. Studies concerning tenosynovial giant cell tumors, GCTB with malignant transformation, case reports, editorials, expert opinions and non-human studies were excluded. The EndNote X9 software (Clarivate Analytics, Philadelphia, PA, USA) was used to remove the duplicate studies from the two databases. Additionally, the references of the initial search were checked for relevance using the same software. Two authors (C.P. and B.C.) screened articles’ titles and abstracts and assessed the eligibility. Disagreements concerning inclusion were discussed with a third author (PG).

### 2.4. Data Extraction and Quality Assessment

Two authors (C.P. and B.C.) extracted the following data and information from each eligible article using Microsoft Excel database: first author’s last name; year of publication; number of patients; number of cases; gender; age; location of the tumor; primary surgery; grade according to Campanacci classification [1]; time interval of recurrence; surgical treatment of recurrent cases; number of re-recurrent cases; surgical treatment of re-recurrent cases; number of third recurrent cases; surgical treatment of third recurrent cases; number of fourth recurrent cases; surgical treatment of fourth recurrent cases; follow-up; Musculoskeletal Tumor Society (MSTS) score [22]; Toronto Extremity Salvage score [23]; and complications.

### 2.5. Data Syntheses and Analysis

Once the data extraction was complete, the data was transcribed into SPSS (IBM Corp., Armonk, NY, USA, released 2017; IBM SPSS Statistics for Windows, Version 25.0. IBM Corp., Armonk, NY, USA) and analyzed. Continuous variables were reported as means with standard deviations or as medians, while categorical variables were reported as proportions. Sub-group column proportions were compared for categorical variables using z-test, and Bonferroni correction was used to adjust for *p* values. All tests were two-sided, and statistical significance was assumed at a *p* value of less than 0.05.

## 3. Results

### 3.1. Study Selection

We initially found 3400 articles from two bibliographic databases based on the search strategy. After screening for duplicates, 1045 articles were excluded. A further 2293 articles were excluded after screening their titles and abstracts. Finally, 62 full-text articles, which referred to the treatment of recurrent GCTB, were retrieved and assessed for eligibility. Of them, 8 were case reports, 5 were case series with less than five patients and 37 included insufficient data (general non-specific information; expert opinions) regarding the study population, the treatment of recurrent GCTB and the final outcome. The selection process ultimately yielded 12 articles suitable for systematic review (Figure 1).

### 3.2. Study Characteristics

Our literature search yielded 12 studies published between 2006 and 2022 that included a total number of 458 patients (49% males) [24,25,26,27,28,29,30,31,32,33,34,35]. The mean age at presentation was 33 years (range: 12–82 years). The initial surgery was IC in 96.8% of cases and EBR in the remaining 3.2%. There was a single recurrence in 365 cases (79.7%), two recurrences in 71 cases (15.5%), three recurrences in 17 cases (3.7%) and four recurrences in 5 cases (1.1%). The mean follow-up after the last surgery was 54.9 months (range: 1–408 months) (Table 1).

### 3.3. Location—Recurrence

The most common locations of the lesion were the proximal tibia and the distal femur, accounting for 26.7% each. Tumors sited in the lower limb and particularly around the knee joint were associated with more frequent tumor recurrence (*p* < 0.001) (Table 2).

### 3.4. Outcomes (Table 3)

#### 3.4.1. Characteristics of Recurrent GCTB

The grade of recurrent GCTB tumors according to Campanacci classification was reported in seven studies containing 225 patients; 19 (8.4%) patients were classified as type 1, 130 (57.8%) as type 2 and 76 (33.8%) as type 3. The mean interval between the initial surgery and the recurrence was reported in five studies, including 133 patients, and it was found to be 23.3 months (range: 2–172 months) (Table 3).

**Table 3 cancers-15-03287-t003:** Treatment of recurrent giant cell tumor of bones.

Author	Year	Recurrence				Second Recurrence				Third Recurrence				Fourth Recurrence			
		NoP	Grade * (NoP)	Interval (Range) (Months)	2nd Intervention	NoP	Grade * (NoP)	Interval (Range) (Months)	3rd Intervention	NoP	Grade * (NoP)	Interval (Range) (Months)	4th Intervention	NoP	Grade * (NoP)	Interval (Range) (Months)	5th Intervention
Von Steyern et al. [32]	2006	19	N/A	17 (3–29)	IC + PMMAc (16) EBR (3)	5	N/A	N/A	IC + PMMAc (3) EBR + RP (1) Amputation (1)	1	N/A	N/A	IC + PMMAc (1)	1	N/A	N/A	EBR + RP (1)
Li et al. [28]	2008	9	N/A	N/A	EBR + Segmental autograft (9)	1	N/A	N/A	N/A	0							
Balke et al. [25]	2009	67	2 (9) 3 (33) N/A (25)	22 (2–172)	IC + Burring + PMMAc (13) IC + PMMAc (11) EBR (11) IC + Burring + H_2_O_2_ + PMMAc (10) IC + BG (9) IC (8) IC + H_2_O_2_ + PMMAc (3) IC + Burring (1) N/A (1)	20	N/A	N/A	IC (8) EBR (5) IC + Burring + PMMAc (3) IC + PMMAc (3) N/A (1)	6	N/A	N/A	EBR (2) IC + Burring + PMMAc (2) N/A (2)	2	N/A	N/A	EBR (1) N/A (1)
Klenke et al. [27]	2011	46	N/A	N/A	IC + PMMAc (14) IC + BG (14) EBR + RP (4) EBR + structural BG (4) EBR + osteoarticular allograft (3) EBR + allograft prosthesis (1) EBR (4) Amputation (2)	6	N/A	24.7 (13–37)	IC (3) EBR (3)	0							
Niu et al. [29]	2012	146	1 (1) 2 (28) 3 (4) N/A (111)	N/A	EBR + RP or segmental allograft (86) IC (60)	12	N/A	N/A	N/A	0							
Wan et al. [34]	2012	27	2 (17) 3 (10)	28.9 (7–97)	EBR + RP (10) IC + PMMAc (17)	3	2 (1) 3(2)	24.7 (12–41)	EBR + RP (2) IC + PMMAc (1)	0							
AlSulaimani et al. [24]	2013	9	1 (1) 2 (3) 3 (4) N/A (1)	32.8 (12–97)	IC + PMMAc (8) IC + Allograft (1)	2	N/A	37	IC + PMMAc (1) IC + PMMAc + radiotherapy (1)	1	N/A	37	IC + Allograft (1)				
Waikakul et al. [33]	2015	8	1 (4) 2 (1) 3 (2) N/A (1)	N/A	IC + Thermoablation ** + ABG (6) IC + Thermoablation ** + PMMAc (1) N/A (1)	3	2 (1) N/A (2)	48	EBR (2) IC + Thermoablation (1)	0							
Takeuchi et al. [30]	2016	103	1 (12) 2 (71) 3 (20)	N/A	IC (85) EBR (17) Amputation (1)	32	N/A	N/A	IC (27) EBR (5)	11	N/A	N/A	IC (9) EBR (1) Amputation (1)	1	N/A	N/A	IC (1)
Deveci et al. [26]	2017	8	1 (1) 2 (1) 3 (3) N/A (3)	N/A	IC + PMMAc (3) IC + PMMAc + Denosumab (1) STR + Denosumab (1) EBR + VFG + Denosumab (1) EBR + RP + Denosumab (1) Denosumab (1)	3	N/A	N/A	IC + PMMAc (3)	3	2 (2) N/A (1)	N/A	Denosumab (2) IC + PMMAc (1)	1	3 (1)	N/A	EBR + RP + Denosum (1)
Zhang et al. [35]	2019	5	N/A	N/A	EBR + Denosumab (3) Denosumab (2)	1	N/A	12	EBR (1)	0							
Tsukamoto et al. [31]	2022	11	N/A	20.6 (3–83)	EBR + RP (4) EBR + allograft (2) IC (3) EBR (1) Amputation (1)	5	N/A	23.6 (3–40)	EBR (4) IC (1)	0							

Abbreviations: ABG: autologous bone grafting, BG: bone graft, EBR: en-bloc resection, FG: fibular grafting, HBG: homologous bone grafting, IC: intralesional curettage, NoP: number of patients, N/A: not applicable, PMMAc: polymethylmethacrylate cement, RP: reconstruction prosthesis, RVTT: reversed vascularized toe transfer, SJ: silicone joint, STR: soft tissue resection, VFG: vascularized fibula graft. * Grade according to Campanacci classification. ** Thermoablation using warm (50 °C) Ringer’s lactate solution for 20 min.

Of the 458 patients with recurrent tumors, the IC was selected in most cases (284 cases, 62.2%, *p* = 0.032). Other treatment modalities included EBR (164 cases, 36%), denosumab injection (4 cases, 0.9%) and amputation (4 cases, 0.9%). In two cases, the treatment was not reported.

The material used to fill the created cavity after IC was reported in seven studies, including 127 patients. PMMAc was introduced in 97 cases (76.4%), while BG (autograft and/or allograft) was used in 30 cases (23.6%) (*p* < 0.001). Among the 127 patients treated with IC and cavity filling, thermoablation using warm (50 °C) Ringer’s lactate solution for 20 min into the cavity was applied in 7 cases (5.5%), and Hydrogen peroxide was utilized in 13 cases (10.2%).

With respect to EBR, the material used for reconstruction of the resected bone was reported in four studies comprising 39 patients. A reconstruction prosthesis was used in 19 patients (48.7%) (*p* < 0.001). Segmental autografts, including vascularized fibula and anterior iliac wing grafts (10 cases, 25.6%), structural BG (4 cases, 10.3%), osteoarticular allograft (3 cases, 7.7%), allograft (2 cases, 5.1%) and allograft prosthesis (1 case, 2.6%) were also applied. Adjunctive denosumab injection was given in five patients after EBR (3.8%).

#### 3.4.2. Second Recurrence

The time interval of second recurrence after the previous surgery was reported in six studies, with a mean value of 28.8 months (range: 3–48 months).

A second recurrence of GCTB was reported in 94 out of 458 recurrent treated cases (20.5%). Of the 94 re-recurrent tumors, 75 (79.8%) had been treated with IC surgery and 18 (19.1%) with EBR (*p* < 0.01). In one patient (1.1%), the treatment method for the recurrent lesion had not been reported.

Subgroup analysis regarding the surgical treatment of the second recurrence of GCTB was available for 440 patients. Intralesional curettage was applied in 276 patients and EBR in 164 patients. A second recurrence was reported in 25.7% (71/276 cases) after IC treatment and in 11% (18/164 cases) after EBR (*p* = 0.012). Following IC, cavity filling with PMMAc showed a lower second recurrence rate (21.6%, 21/97 cases) than BG (43.3%, 13/30 cases) (*p* < 0.001). A respective subgroup analysis for EBR could not be achieved.

Details regarding the surgical treatment were reported in 11 studies, including 79 out of 94 recurrent cases. Intralesional curettage was performed in 55 patients (69.6%) and EBR in 23 patients (29.1%) (*p* < 0.001). In one case, an amputation was performed (1.3%). The PMMAc was exclusively used in all 15 IC cases where the filling material was reported. Similarly, a reconstruction prosthesis was applied in all three cases and referred to combined EBR procedures.

Adjunctive radiotherapy was used in one patient after IC and PMMAc filling and thermoablation in one case after IC alone.

#### 3.4.3. Third Recurrence

The time interval after the previous surgery was reported in one study, and it was 37 months.

Twenty-two out of the ninety-four patients (23.4%) with a re-recurrent GCTB experienced a third recurrence. All patients were treated with IC. The overall rate of tumor relapse was 46.8% after IC. No LR was reported after the previous EBR.

The treatment method of the third LC was reported in 21 out of 22 patients. Fourteen patients were treated with IC (63.5%, *p* < 0.001), four with EBR (18.2%), two with denosumab injections only (9.1%) and one with amputation (6.2%).

#### 3.4.4. Fourth Recurrence

Five out of fourteen GCTB treated cases for the third recurrence (35.7%) developed another recurrence. All five patients had undergone a previous IC surgery.

The fourth tumor recurrence was treated with EBR (three cases) or with IC in one patient. The treatment method of the last patient was not reported. In two out of the three cases managed with EBR, a reconstruction prosthesis alone (one patient) or with adjunctive denosumab injection (one patient) was subsequently applied. No further recurrence was reported.

### 3.5. Patient-Reported Outcome Measures (PROMs)

Regarding the PROMs, the final MSTS score at the time of the latest follow-up was reported in four studies, including 69 patients. The overall mean score was 82.7 (range: 25–100). The increased number of recurrences was correlated with a lower score (*p* = 0.023). Surgical treatment of tumor recurrence was associated with a mean score of 77.9 after IC (24 cases) and 87 after EBR (45 cases) (*p* = 0.116).

In the subgroup analysis, the MSTS score was higher when IC was combined with PMMAc instead of BG (84.2 vs. 34.2, *p* < 0.001). Additionally, structural allografts after EBR were associated with improved functional outcomes compared with reconstruction prostheses (89.8 and 62.5, respectively, *p* = 0.028). (Table 4)

### 3.6. Complications

Neurological deficit and neuropathic pain were recorded in two sacral lesions treated with IC, thermoablation and BG. Infection (three cases) and joint stiffness (one case) were also complicated EBR procedures.

## 4. Discussion

The most important finding of this systematic review is that one-fifth of patients (20.5%) treated for recurrent GCTB may experience a second recurrence. Furthermore, a third recurrence of similar incidence may also appear after surgical treatment of the second recurrence. Surgical treatment with IC, particularly when combined with BG instead of PMMAc, was associated with a higher second recurrence rate compared to EBR. However, it is interesting that the overall functional outcome was satisfactory and almost equal between IC and EBR. The increased number of surgical interventions and the IC with autograft filling were correlated with the worst outcome.

A second LR rate can vary from 11% to 55%, depending on the therapeutic modality utilized. This rate is equal to the frequency of the first LR [12,34]. Niu et al. [29], studying 146 patients with recurrent GCTB, found that the overall rate of local second recurrence was not higher than the first recurrence. They also reported that EBR and reconstruction with a prosthesis or segmental allograft were applied more frequently after tumor recurrence. In their comparative study between IC and EBR treatments in both primary and recurrent GCTB cases, Becker et al. [12] reported similar LR rates. Additionally, they found that recurrent lesions treated with IC yielded a superior rate of recurrence-free outcomes compared with primary tumors treated with the same procedure. This was attributed to the possibly more aggressive and meticulous surgical debridement of recurrent tumors.

After IC, the remaining osseous cavity is usually filled with BG or PMMAc [36]. In the systematic review of Vaishya et al. [37], the use of bone cement was associated with a lower recurrence rate compared to BG after extended IC in primary cases. In recurrent lesions, the filling of the cavity with PMMAc after IC also yields significantly lower recurrence rates [34]. Our results also pointed out that the use of BG is associated with a higher recurrence rate than PMMAc during IC surgery. After the application of PMMA, a rim of cellular necrosis beyond that of burring has been identified in histological studies [38]. The toxic action of monomer and the heat produced during PMMAc polymerization can facilitate the elimination of the residual tumor cells [39,40].

Adjuvant therapies such as denosumab may also have a positive impact on the outcome of GCTB treatment [41]. Denosumab is a monoclonal antibody used in cases of unresectable or huge recurrent GCTB for reducing the tumor size [42]. Its application has been occasionally restricted due to possible systematic complications, including osteonecrosis in the jaw, arthralgia, fatigue, headache, anemia and hypercalcemia [43]. Tsukamoto et al. [44] administrated denosumab before curettage of GCTB in patients suffering from large extraosseous lesions, pathological fractures or minimal residual periarticular and subchondral bone stock. They identified that preoperative denosumab application was associated with a lower recurrence rate, improved functional score and higher joint preservation rate. In the systematic review of Jamshidi et al. [45], denosumab was associated with 2% of second recurrence and minimal complications. The authors suggested denosumab use as an alternative along with surgery in complex cases or when surgery could not be applied. In the retrospective study of Deveci et al. [26], none of the eight patients with recurrent GCTB treated with adjunctive subcutaneous denosumab had another recurrence or experienced any complication associated with its use. According to the current evidence, denosumab not only can reduce the extent of surgical resection but can also be used as monotherapy for unresectable and recurrent lesions, particularly in medically compromised patients [46,47].

The MSTS score is a commonly used evaluation system for the assessment of the postoperative outcome of GCTB therapy [48,49,50]. It is generally accepted that the EBR of GCTB is associated with the worst functional outcome [14]. Wan et al. [34] studied 27 patients operated on for recurrent GCTB. They found that the MSTS score was significantly lower in patients treated with wide EBR than those treated with IC. Regarding the impact of the number of LR on the functional outcome, AlSulaimani et al. [24] reported that MSTS score was not affected after multiple LR. On the contrary, our results indicate that the number of recurrences and the respective procedures were inversely correlated with a good functional outcome. Moreover, the treatment of LR with IC and autologous bone grafting was associated with the worst functional score.

Another important factor affecting the outcome of the treatment of GCTB is the preservation of the adjacent joint. In their study of 45 patients with recurrent GCTB, McGough et al. [51] found that incomplete initial surgery, more than 6 months of delay in the diagnosis of recurrence and subchondral LR might predispose to failure of joint preservation and mobility. Takeuchi et al. [30] advocated that repetitive recurrence was not a risk factor for sacrificing the adjacent joint, as they reported an incidence of 76.6% of joint preservation in patients who encountered multiple recurrences after IC treatment. We found that multiple recurrences could jeopardize the affected joint and limb integrity as a gradual increase in amputation rate after treatment of first (0.9%), second (1.3%) and third (6.2%) tumor recurrence was identified.

This systematic review has inherent limitations that are mostly related to the type of studies included. Specifically, there was inconsistency among the studies with respect to the adjuvant treatment modalities used, especially after IC. Secondly, the authors rarely reported if marginal or wide excision was the aim of surgery and if histologically clear margins were achieved after EBR. Therefore, a subgroup analysis based on the surgical margins of resection could not be achieved. Thirdly, analysis of GCTB cases suffering from metastasis, malignant transformation and joint resection were not performed. Finally, we could not associate the relationship between patient and tumor demographics with a second recurrence rate due to the heterogeneity of the available data.

## 5. Conclusions

The management of recurrent GCTB is challenging, as relapse is correlated with impaired outcomes. This review confirmed the high second recurrence rate of recurrent GCTB. The chosen treatment method seems to largely affect the incidence of second recurrence as IC has been associated with more failures compared to EBR. However, the combination of IC and PMMAc seems to decrease the second recurrence of GCTB, as it yields both a relatively low LR rate and satisfactory PROMs. Due to the paucity of data, further research with randomized control trials is required in order to enable the extraction of more robust conclusions regarding the best management of recurrent GCTB.

## Figures and Tables

**Figure 1 cancers-15-03287-f001:**
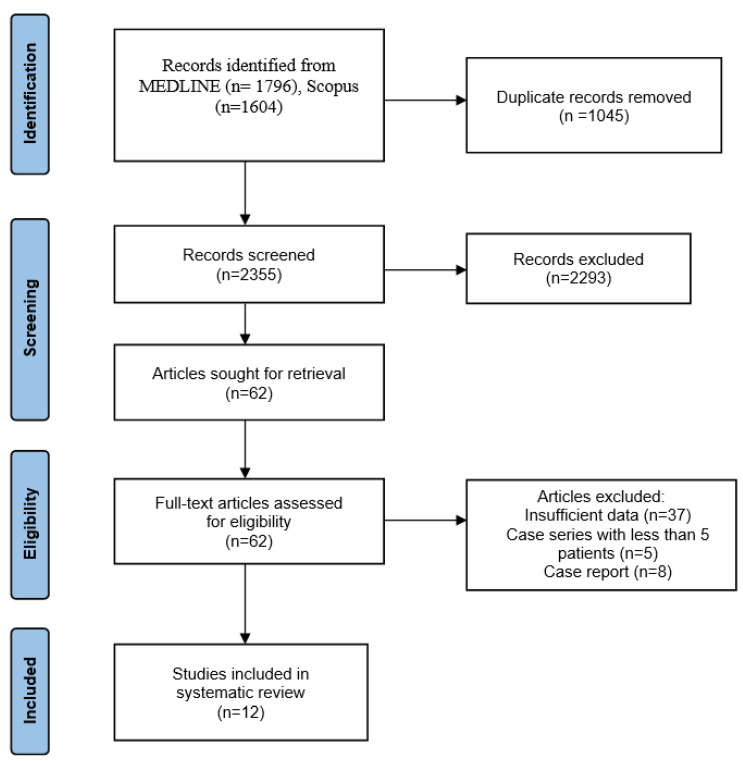
PRISMA flow diagram with research results.

**Table 1 cancers-15-03287-t001:** Basic characteristics of the included studies about recurrent giant cell tumor of the bone.

Author	Year	NoP	Mean Age at Presentation (Range) (Years)	Gender (Male/Female)	Location (NoP)	Primary Surgery	Number of Recurrences (NoP)	Last Follow-Up (Range) (Months)
Von Steyern et al. [32]	2006	19	30.8 (18–73)	10/9	Distal femur (6) Proximal tibia (5) Distal radius (3) Proximal femur (2) Proximal humerus (1) Proximal fibula (1) Distal tibia (1)	IC + PMMAc (19)	1 (14) 2 (4) 4 (1)	53 (3–128)
Li et al. [28]	2008	9	28.8 (21–48)	5/4	Proximal tibia (9)	IC + BG (7) IC + PMMAc (1) IC + Artificial bone (1)	1 (8) 2 (1)	118.7 (60–153)
Balke et al. [25]	2009	67	29.6 (16–63)	32/35	Distal femur (20) Proximal tibia (11) Distal radius (8) Proximal femur (5) Distal tibia (5) Ribs (3) Distal humerus (2) Sacrum (2) Proximal fibula (2) Foot (2) Scapula (1) Proximal humerus (1) Proximal ulna (1) Distal ulna (1) Hand-metacarpal (1) Patella (1) Distal fibula (1)	IC (32) IC + PMMAc (31) IR (3) N/A (1)	1 (47) 2 (14) 3 (4) 4 (2)	45.3 (1–209)
Klenke et al. [27]	2012	46	31.1	21/25	Distal femur (11) Proximal tibia (9) Distal radius (7) Proximal humerus (5) Proximal femur (5) Distal tibia (5) Distal humerus (2) Proximal fibula (2)	N/A (46)	1 (40) 2 (6)	134 (37–337)
Niu et al. [29]	2012	146	N/A	N/A	Distal femur (13) Proximal tibia (9) Proximal femur (8) Distal radius (2) Proximal humerus (1) Other (2) N/A (111)	IC (33) EBR (2) N/A (111)	1 (134) 2 (12)	N/A
Wan et al. [34]	2012	27	38 (22–58)	13/14	Proximal tibia (12) Distal femur (7) Proximal femur (3) Proximal humerus (2) Distal radius (1) Distal metacarpal (1) Distal tibia (1)	IC + PMMAc (14) IC + Allograft (8) IC + ABG (5)	1 (24) 2 (3)	46.8 (24–78)
AlSulaimani et al. [24]	2013	9	37.8 (21–63)	3/6	Distal tibia (9)	IC + ABG (2) IC + Autograft + Allograft (2) IC + Allograft + PMMAc (1) IC + Allograft + PMMAc + Liquid nitrogen (1) IC+ Phenol + Allograft (1) IC + H_2_O_2_ + Allograft (1) IC + PMMAc (1)	1 (7) 2 (1) 3 (1)	83.3 (25–229)
Waikakul et al. [33]	2015	8	28 (12–50)	4/4	Radius (2) Sacrum (2) Clavicle (1) Hand (1) Distal femur (1) Distal tibia (1)	N/A	1 (5) 2 (3)	48 (48)
Takeuchi et al. [30]	2016	103	34 (12–82)	53/50	Proximal tibia (37) Distal femur (28) Distal radius (12) Proximal femur (9) Proximal humerus (6) Distal ulna (3) Distal tibia (3) Hand (2) Patella (1) Proximal fibula (1) Foot (1)	IC: 98 EBR: 5	1 (71) 2 (21) 3 (10) 4 (1)	114 (11–408)
Deveci et al. [26]	2017	8	37.5 (26–50)	4/4	Distal femur (4) Distal radius (2) Proximal humerus (1) Proximal fibula (1)	IC + PMMAc (4) IC + BG (3) EBR (1)	1 (5) 3 (2) 4 (1)	N/A
Zhang et al. [35]	2019	5	40.2 (19–67)	¼	Sacrum (2) Proximal humerus (1) Distal radius (1) Distal tibia (1)	IC + Denosumab (2) N/A (3)	1 (4) 2 (1)	30 (13–45)
Tsukamoto et al. [31]	2022	11	40.3 (22–71)	5/6	Distal radius (4) Proximal femur (2) Distal femur (2) Proximal humerus (1) Distal ulna (1) Hand (1)	EBR + Allograft (5) EBR + RP (4) EBR (1) EBR + FG (1)	1 (6) 2 (5)	70.1 (41–124)

Abbreviations: ABG: autologous bone grafting, BG: bone graft, EBR: en-bloc resection, FG: fibular grafting, IC: intralesional curettage, IR: intralesional resection, N/A: not applicable, NoP: number of patients, PMMAc: polymethylmethacrylate cement.

**Table 2 cancers-15-03287-t002:** Site of recurrent giant cell tumors.

Site	Number of Patients	Rate (%)
Proximal tibia	92	26.7
Distal femur	92	26.7
Distal radius	40	11.6
Proximal femur	34	9.9
Distal tibia	26	7.5
Proximal humerus	19	5.5
Hand-metacarpal	6	1.7
Sacrum	6	1.7
Distal ulna	5	1.5
Proximal fibula	7	2
Ribs	3	0.9
Foot	3	0.9
Distal humerus	4	1.2
Patella	2	0.5
Radius (unspecified)	2	0.5
Clavicle	1	0.3
Scapula	1	0.3
Proximal ulna	1	0.3
Distal fibula	1	0.3

**Table 4 cancers-15-03287-t004:** Reported functional scores at the last follow-up.

Author	Year	Number of Patients Evaluated	Musculoskeletal Tumor Society Score (Range)
Li et al. [28]	2008	8	92.9 (80–100)
Wan et al. [34]	2012	27	85.6 (50–97)
AlSulaimani et al. [24]	2013	5	32.6 (25–35)
Tsukamoto et al. [31]	2022	29	85.8 (57–100)

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
