# Peer review of "Treatment of Recurrent Giant Cell Tumor of Bones: A Systematic Review"

_cancers, 2023, doi:10.3390/cancers15133287_

Round 1

Reviewer 1 Report

Dear Authors

Following the WHO Bone and Soft tissue tumors classification, the giant cell tumor of bone is an intermediate tumor, i.e., neither benign nor malignant. Even if you consider it a benign tumor, the authors should at least comment on the WHO classification.

Is it possible to separate axial lesions (sacral, ribs, etc.) from limb lesions regarding the rate and number of recurrences? 

Also, the information provided would be helpful if the authors separate proximal tumors (proximal femur, proximal humerus, pelvic non-sacral) from distal lesions.

I needed clarification about the information regarding the use of denosumab. The Jamshidi systematic review does not indicate that denosumab decreases the local recurrence rate. So, the authors should provide better data about denosumab use and the rate of 1st, 2nd, 3rd, and 4th recurrence.

Reviewer 2 Report

Thank you for the invitation to review this paper. These are my comments.

I would ask the authors to clarify the EBR. What are the margins of EBR? Clarify the type of resection and reconstruction after resection. Resection usually refers to wide margins excision for malignant tumors. Maybe consider replace with excision if not related to wide margins.

Please discuss the role of denosumab for (re-)recurrences. Refer to current literature.

Table 1 does not give interesting information; I recommend to merge with Tables 3and 4.

Thank you for the invitation to review this paper. These are my comments.

I would ask the authors to clarify the EBR. What are the margins of EBR? Clarify the type of resection and reconstruction after resection. Resection usually refers to wide margins excision for malignant tumors. Maybe consider replace with excision if not related to wide margins.

Please discuss the role of denosumab for (re-)recurrences. Refer to current literature.

Table 1 does not give interesting information; I recommend to merge with Tables 3and 4.

Author Response

   Dear Editor,

   We would like to thank you for accepting to reconsider our manuscript titled: “Treatment of Recurrent Giant Cell Tumor of Bones: A Systematic Review” (Manuscript ID: cancers-2424244) for publication in the special issue “Giant-Cell-Containing Tumors of Bone—New Insights into Pathobiology, the Clinical Setting and Targeted Therapies” of Cancers Journal.

   We would also like to thank the reviewers for their insightful comments. All raised points have been addressed and the manuscript has been revised and edited according to their suggestions and instructions. All text changes in the manuscript have been highlighted. For reviewing purposes, the comments have been addressed one by one.

   In more detail:

Reviewer 1:

Comment: “I would ask the authors to clarify the EBR. What are the margins of EBR? Clarify the type of resection and reconstruction after resection. Resection usually refers to wide margins excision for malignant tumors. Maybe consider replace with excision if not related to wide margins.”

Reply: Thank you for the comment. In the majority of studies no definite information regarding the specific margins of en-bloc resection (EBR) has been provided. The authors rarely reported if marginal or wide excision was the aim of surgery and if histologically clear margins were achieved. A relevant comment has been added in Discussion section (line 287-290). As in almost all the included studies, the term “en-block resection” (EBR) was reported, we also used the EBR to describe the option of complete tumor resection.

Regarding the reconstruction method following EBR, the selected approach was highly associated with tumor location, available bone stock and integrity of articular surfaces. A bone graft was used in case of joint preservation or joint fusion. Alternatively, a reconstruction prosthesis was applied to restore joint mobility and function. This information is presented in Introduction section (lines 41-46).

Comment: “Please discuss the role of denosumab for (re-)recurrences. Refer to current literature.”

Reply: Thank you for the comment. We have added new information regarding the role of denosumab in the treatment of recurrent GCTB and we have rewritten the relevant paragraph. (lines 248 – 264)

Comment: “Table 1 does not give interesting information; I recommend to merge with Tables 3 and 4.”

Reply: Thank you for the comment. Table 1 (“Basic characteristics of the included studies about recurrent giant cell tumor of the bone”) contains demographic and general information of the included studies and the treatment choice of the primary GCTB. The tables 3 and 4 provide details about the treatment method of the recurrent or multiple-recurrent tumors and the final functional outcomes. Therefore, we believe that by merging all the relevant data, the new Table 1 (old Tables 1, 3, 4) may become very  large and confusing.

Reviewer 3 Report

The review by Pitsilos et al was a well organized manuscript that described the current state of treatment of recurrent giant cell tumors (GCT) of bone. In it, the authors sought to summarize the collective experience in the literature as it relates to current therapy of GCT of bone.

There are several issues that the authors should address prior to acceptance by the journal:

1. The authors identified > 2000 unique articles on treatment of recurrent GCT of bone. Yet only 12 were included in the final analysis. There was a consort diagram but no specific mention of why so few articles made it in to the final analysis. Under section 3.1, the authors mentioned "exclusion criteria" that I could not easily locate. This small number of articles out of a potentially larger set may be a bias in and of itself. The authors should include also the time period for the literature review.

2. Under section 3.4.2, the numbers do not add up for me in terms of the numbers of patients treated. Perhaps the authors could reword the section.

3. The term "re-recurrence" is used multiple times throughout the manuscript. I would be more specific about the recurrence that the author is talking about. For example, "first recurrence", "second recurrence", etc.

4. In section 3.4.3, the abbreviation "LC" is used. What is this an abbreviation for?

5. In table 4, I would specify that it is the "Musculoskeletal Tumor Society Score" in the fourth column of the table.

6. In lines 234-235 in the discussion section, the authors appear to attribute causality without presenting data for the mechanism of PMMAc and tumor killing. I am not sure this is needed.

There are minor edits that might improve the readability of the manuscript. Overall it was excellent.

Author Response

   Dear Editor,

   We would like to thank you for accepting to reconsider our manuscript titled: “Treatment of Recurrent Giant Cell Tumor of Bones: A Systematic Review” (Manuscript ID: cancers-2424244) for publication in the special issue “Giant-Cell-Containing Tumors of Bone—New Insights into Pathobiology, the Clinical Setting and Targeted Therapies” of Cancers Journal.

   We would also like to thank the reviewers for their insightful comments. All raised points have been addressed and the manuscript has been revised and edited according to their suggestions and instructions. All text changes in the manuscript have been highlighted. For reviewing purposes, the comments have been addressed one by one.

   In more detail:

Reviewer 2

Comment: “The authors identified > 2000 unique articles on treatment of recurrent GCT of bone. Yet only 12 were included in the final analysis. There was a consort diagram but no specific mention of why so few articles made it in to the final analysis. Under section 3.1, the authors mentioned "exclusion criteria" that I could not easily locate. This small number of articles out of a potentially larger set may be a bias in and of itself. The authors should include also the time period for the literature review.”

Reply: Thank you for both comments. The purpose of this systematic review was to collect and analyze information about the treatment of recurrent GCTB. The initial search was based on general terms in order to scrutinize the literature about related articles. However,  only 62 articles referred to treatment of recurrent GCTB according to their Titles and Abstracts. Of them, 8 were case reports, 5 were case series with less than 5 patients and 37 included insufficient data (general non-specific information, experts opinion) regarding the study population, the treatment of recurrent GCTB and the final outcome. So, the final selection included only 12 studies. We have added this information in “Study Selection” section of the article. (lines 112-117)

In addition, and according to your comment, the time period for the literature review has been clearly presented in line 76.

Comment: “Under section 3.4.2, the numbers do not add up for me in terms of the numbers of patients treated. Perhaps the authors could reword the section.”

Reply: We agree with this comment. This section has been rewritten in order the numbers referred to patients who treated for tumor recurrence  to be clearer and more understandable. (lines 167-170)

Comment: “The term "re-recurrence" is used multiple times throughout the manuscript. I would be more specific about the recurrence that the author is talking about. For example, "first recurrence", "second recurrence", etc.”

Reply: We have replaced the term “re-recurrence” with “second recurrence”. Moreover, and according to your comment, we preserved the terms “third recurrence” and “fourth recurrence”.

Comment: “In section 3.4.3, the abbreviation "LC" is used. What is this an abbreviation for?”

Reply:  We apologize for any inconsistency. The correct abbreviation “LR” instead of LC is now used.

Comment: “In table 4, I would specify that it is the "Musculoskeletal Tumor Society Score" in the fourth column of the table.”

Reply: Thank you for the comment. The word “Score” after “Musculoskeletal Tumor Society” was added.

Comment: “In lines 234-235 in the discussion section, the authors appear to attribute causality without presenting data for the mechanism of PMMAc and tumor killing. I am not sure this is needed”

Reply: There is evidence that temperature elevation and increased exposure time may induce  significant tissue necrosis at the margin of the tumor site (ref 36).  Therefore, the monomer of PMMA may have a possible thermal and cytotoxic effect  and contribute to inhibition of tumor recurrence. This information along with the appropriate references have been added in the Discussion section and the relevant paragraph has been modified accordingly. (lines 244 to 247).

Round 2

Reviewer 2 Report

Accept

Moderate editing is necessary